METHODS

# A simple method to efficiently generate structural variation in plants

Lindsey L. Bechen[1], Naiyara Ahsan [2,3], Alefiyah Bahrainwala[2], Mary Gehring [1,4,5,6*], Prasad R. V. Satyaki [1,2,3*]

1 Whitehead Institute for Biomedical Research, Cambridge, Massachusetts, United States of America, 2 Department of Biological Sciences, University of Toronto Scarborough, Toronto, Ontario, Canada, 3 Department of Cell and Systems Biology, University of Toronto, Toronto, Ontario, Canada, 4 Department of Biology, Massachusetts Institute of Technology, Cambridge, Massachusetts, United States of America, 5 Department of Biological Engineering, Massachusetts Institute of Technology, Cambridge, Massachusetts, United States of America, 6 Howard Hughes Medical Institute, Whitehead Institute for Biomedical Research, Cambridge, Massachusetts, United States of America

* mgehring@wi.mit.edu (MG); satyaki.rajavasireddy@utoronto.ca (PRVS)

## Abstract

Phenotypic variation is essential for the selection of new traits of interest. Structural variants, consisting of deletions, duplications, inversions, and translocations, have greater potential for phenotypic consequences than single nucleotide variants. Pan-genome studies have highlighted the importance of structural variation in the evolution and selection of novel traits. Here, we describe a simple method to induce structural variation in plants. We demonstrate that a short period of growth on the topoisomerase II inhibitor etoposide induces heritable structural variation and altered phenotypes in *Arabidopsis thaliana* at high frequency. Using long-read sequencing and genetic analyses, we identified deletions and inversions underlying semi-dominant and recessive phenotypes. This method requires minimal resources, is potentially applicable to any plant species, and can replace irradiation as a source of induced large-effect structural variation.

## Author summary

Improvement of crop species relies on selecting from amongst existing natural variation in the traits a species exhibits. However, the standing diversity of a given crop species is unlikely to encompass all desirable possibilities. Crop breeders therefore often induce new traits by changing DNA sequence at random locations. These changes may be either at the scale of single nucleotides in the genome or at the scale of tens to millions of bases, depending on the method used. Inducing large changes to DNA is desirable because such changes are more likely to create new traits. However, current methods to induce such changes require a radiation source and thus may not be accessible to crop breeders

**Data availability statement:** Short-read DNA sequencing, long-read DNA sequencing, and mRNA-seq data are available in GEO records GSE284455, GSE284456 and GSE284457.

**Funding:** This research was funded by grants awarded to MG by The Abdul Latif Jameel Water and Food Systems Lab (https://jwafs.mit.edu/), The Professor Amar G. Bose Research Grant Program (https://bosefellows.mit.edu/), and the Vincent J Ryan Orphan Plant Project. MG is a Howard Hughes Medical Institute Investigator. PRVS's work at University of Toronto was funded by Natural Sciences and Engineering Research Council RGPIN-2024-05508. The funders had no role in study design, data collection and analysis, decision to publish, or preparation of the manuscript.

**Competing interests:** I have read the journal's policy and the authors of this manuscript have the following competing interests: MG and PRVS are listed on a patent application filed by the Whitehead Institute that is related to the methods/findings described in this manuscript.

world-wide or are tedious to apply. Here we describe a new method that repurposes the well-studied cancer drug etoposide to induce large genetic changes to plant genomes and demonstrate its efficacy in *Arabidopsis thaliana*. This method is simple, affordable, and potentially applicable to a wide variety of plant species.

## Introduction

Genomic structural variations (SVs) – insertions, deletions, duplications, translocations and inversions – are an important source of genetic and phenotypic novelty. Structural variants create genetic novelty by multiple means, including fusing the coding regions of two genes, altering cis-regulatory environments and gene expression patterns, and by altering or suppressing recombination, among other mechanisms. In addition, deletions and duplications alter gene or chromatin dosage [1–3]. Novel structural variants have larger phenotypic effects per mutation than single nucleotide variants [4] and have been selected for when they underlie traits beneficial to the organism or desirable to agriculturists [5,6].

Recent pan-genome sequencing of multiple plant species has underscored the versatility and significance of structural variation to phenotypic variation, modern crop traits, and crop improvement. Lower seed sets in watermelons and grapes, important for the development of popular "seedless" varieties, have been linked to large inversions that cause meiotic defects [7,8]. SVs have also been linked to the domestication of broomcorn millet [9] and sorghum [10]. In pearl millets, gain of heat tolerance has been linked to SVs at multiple loci [11]. SVs are also associated with variation in immunity and plant defense responses [12–14]. In Arabidopsis, a multi-enzyme pathway synthesizes the glucosinolate family of defense compounds. SVs that either delete or fuse enzyme-encoding paralogs modify enzymatic pathways and are responsible for ecotype-specific variation in glucosinolates [15]. Studies of *Oryza sativa* provide numerous examples of how SVs influence phenotypes. Change in copy number of *VIL1* is associated with changes to flowering time and grain number variation [16]. In addition, duplications of the *KALA4* gene – a regulator of the anthocyanin biosynthesis pathway – create a novel cis-regulatory region that promotes ectopic expression of anthocyanins and causes a "black pericarp" phenotype [17]. In peaches, fruit flesh color around the stone and fruit shape are dependent on a deletion and a megabase scale inversion [18]. Modern varieties of sweet corn were created by selecting for an inversion that creates a loss-of-function mutation in the *shrunken-2* gene, which encodes the first enzyme of the starch biosynthesis pathway [19,20]. In tomatoes, a duplication of a cytochrome P450 gene is linked to increased fruit weight [5].

Extant crop genetic diversity is unlikely to provide all possible SV diversity that could impact crop improvement. Induced structural variation can therefore provide novel germplasm for breeding programs, information that can be used for targeted CRISPR-mediated restructuring of crop genomes, and novel genetic variants for understanding fundamental aspects of plant biology. The most common method

employed to induce random structural variation is exposure to ionizing radiation, including X-rays, gamma irradiation, and heavy ions [21–26]. Since the 1960's, irradiation has been used to induce structural variation in model organisms and in numerous crops including, more recently, rapeseed [27], wheat [28], cotton [29], rice [23,30], poplar [22,31], soybean [32,33], and buckwheat [34]. Populations carrying such structural variation have been used for the isolation of mutations that abolish hybrid sterility in rice [35] and the creation of seeds differing in oil composition [21,27].

Although radiation-induced structural variation remains an important element of modern plant breeding, there are logistical challenges. In contrast to the induction of single nucleotide variants that researchers can conveniently pursue in their own labs using ethyl methanesulfonate (EMS) or sodium azide [36], the induction of structural variants requires access to radiation sources. Radiation sources are heavily regulated and access to nuclear reactors, particle accelerators, and other sources of radiation is a bottleneck in the creation of structural variant libraries [26]. Additionally, application of irradiation can be tedious. For example, the creation of a mutant population in poplar required irradiation of collected and dried pollen [22]. One alternative is the extended TAQing approach, in which a transgene expresses a restriction enzyme to induce conditional double-stranded DNA breaks, which then leads to shuffling of the plant genome [37]. However, this approach can be challenging as the commercial varieties of many crops are not easily transformed.

To surmount the current challenges of creating SVs in plants, we sought a simple method that could be applied to any plant species of interest to induce SVs with high frequency. We drew inspiration from the ubiquity and ease of EMS chemical mutagenesis to induce single nucleotide variants. One possible class of mutagens for SV induction is DNA topoisomerase II (Topo II) inhibitors. Topo II relaxes torsional stress from DNA supercoiling generated during DNA replication or transcription by transiently breaking both strands and then ligating them after passing a DNA segment through the break. Between strand breakage and ligation, Topo II is covalently linked to DNA via a tyrosine residue, forming a topoisomerase cleavage complex [38]. This complex is stabilized by the inhibitor etoposide. A collision between covalently-linked Topo II and DNA polymerases during DNA replication, or with RNA polymerases during transcription, leads to removal of the Topo II enzyme, which results in the generation of double-stranded breaks (DSBs) [39–42]. The imprecise repair of DSBs leads to genomic rearrangements and structural variation in mouse spermatocytes, fibroblasts, and in human cells [43–45]. Previously, it was shown that treatment with etoposide impacts genome stability and inhibits plant growth in *Arabidopsis thaliana*, *Allium cepa*, and *Lathyrus sativus* [46–49] and causes chromosomal fragmentation during meiosis in Arabidopsis [48]. However, its potential as a mutagen that can induce structural variation in plants has not been investigated. Here, we provide proof-of-principle that the chemotherapeutic drug etoposide efficiently generates novel genomic structural variation in *Arabidopsis thaliana*.

## Description of the method

In brief, the procedure was to germinate and grow seeds on media containing etoposide for 2–3 weeks until multiple true leaves developed. Etoposide-treated seedlings were then transferred to soil for the remainder of the life-cycle (Fig 1A). Progeny of these plants were screened for phenotypes of interest (Figs 1B–F, S1).

Etoposide (Abcam AB120227) stock was created by dissolving etoposide salt in DMSO to obtain a 100 mM solution. This stock was added to 0.5x Murashige and Skoog (MS) media supplemented with 1% sucrose and Phytoagar to final etoposide concentrations in the 0–640 µM range. Etoposide solubility in aqueous solutions decreased beyond concentrations of 160 µM. We found that the efficiency of etoposide varied by batch and by the species to which etoposide was applied. It is therefore prudent to identify the highest concentration of etoposide resulting in reduced seedling growth without lethality (Fig 1A). Wild-type Col-0 seeds were sterilized by incubating in 2% (v/v) plant preservative mixture (Plant Cell Technologies) for three days at 4°C and sowed on MS media plates supplemented with sucrose and differing concentrations of etoposide (0 µM i.e. DMSO only, 20 µM, 40 µM, 80 µM, 160 µM, 320 µM, or 640 µM) for up to two weeks (Fig 1A). Plants grown on DMSO alone or 20 µM etoposide showed no observable growth defects (Fig 1A). Two-week-old seedlings grown on 40 µM etoposide exhibited gnarled leaves but no

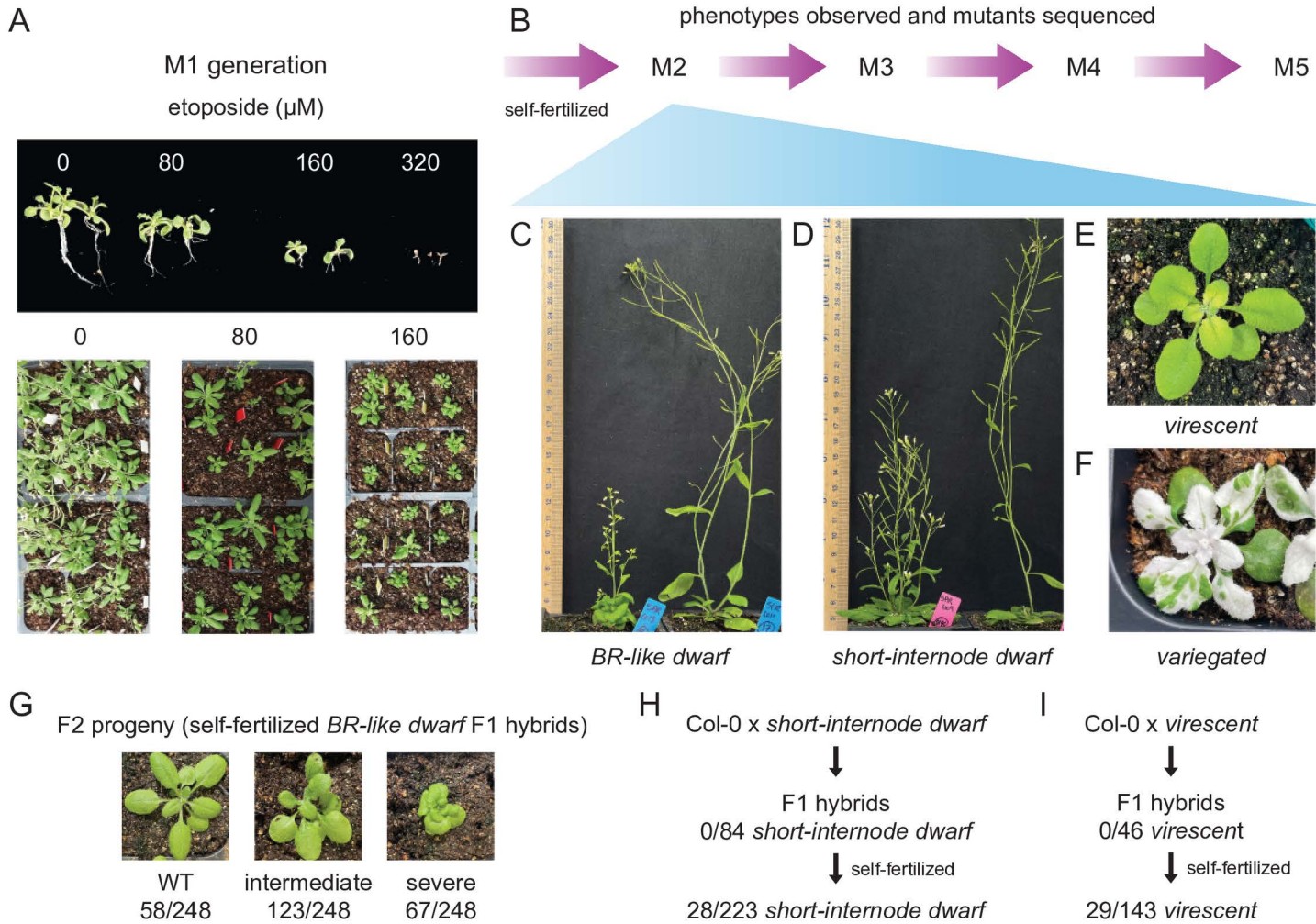

**Fig 1. Etoposide induces novel heritable phenotypes.** (A) Arabidopsis seeds germinated on increasing concentrations of etoposide show a dose-responsive growth defect (upper panel). Seedlings grown on 320 μM etoposide did not exhibit true leaves. This generation is referred to as M1. Growth defects persisted after transplantation to soil (lower panel). (B) M1 was self-fertilized to give rise to M2. Subsequent rounds of self-fertilization produced M3, M4, and M5 generations. A visual examination of M2 generation identified multiple phenotypes, including: (C) a *brassinosteroid-like dwarf* phenotype, (D) a fertile dwarf with short internodes, (E) a *virescent* phenotype with yellowish leaves, and (F) a *variegated* phenotype. (G-I) describe crosses to the wild-type and subsequent selfing to identify the nature of inheritance of the mutant phenotypes and the number of contributing loci. (G) The *BR-like dwarf* phenotype is incompletely dominant and displays three phenotypic classes. F2 segregation data suggests that this phenotype is caused by mutation at a single locus. (H) The *short-internode dwarf* phenotype is recessive and is likely caused by more than one locus. (I) The *virescent* phenotype is recessive and caused by a single locus.

differences in size. Seedlings grown on 80 μM etoposide showed marginally reduced growth (Fig 1A). Of the seedlings grown on 160 μM of etoposide, less than half (75/184) developed true leaves. In addition, these plants displayed stunted root growth (Fig 1A). Seedlings grown on 320 μM or 640 μM etoposide showed little root or shoot growth and exhibited high seedling lethality. Seedlings grown on DMSO only, 80 μM, and 160 μM etoposide (referred to as M1 plants) were transplanted to soil and compared at maturity. Those exposed to 160 μM etoposide exhibited significantly more abnormal phenotypes than DMSO only or 80 μM etoposide plants, including loss of apical dominance, gnarled leaves, reduced plant size, seed abortion, and lower seed number at maturity (Fig 1A). M2 seeds were collected from mature M1 plants treated with 160 μM etoposide.

## Verification and comparison

### Treatment with etoposide results in mutants with a variety of phenotypes

To test if M1 plants produced progeny with mutant phenotypes, we scored for six visible phenotypes—dwarfism, loss of apical dominance, seed size or abortion, leaf shape or size, flowering time, and leaf pigmentation—among the progeny (M2 generation) of plants treated with 160 µM etoposide. Of M2 progeny derived from 42 different M1 parents, 29 lines exhibited at least one obvious visible phenotype (Figs 1B–F, S1, S1 Table). These phenotypes included: variegated or albino plants, altered flowering time, altered leaf shape and size, sterility or seed abortion, and dwarfism (S1 Fig, S1 Table). The large proportion of plants showing visible phenotypes suggested that etoposide could be an excellent mutagen for efficiently creating large-effect mutations.

### Etoposide treatment induces a spectrum of structural variation types

To characterize the molecular nature of etoposide-induced mutations, we employed Illumina short-read sequencing technology to sequence the genomes of 32 M2 or M3 progeny of 15 etoposide-treated M1 plants (S2 Table). To capture the full mutational spectrum, we included individuals with and without a scored mutant phenotype (see S3 Table for relationships). To enhance the accuracy of our structural variation (SV) and other mutation calls, we also identified mutations segregating in our laboratory Col-0 population by sequencing four M2 plants descended from four control M1 lines treated only with DMSO.

To identify etoposide-induced SVs, we first adapted a computational approach that uses genome-wide coverage to identify large segmental deletions or duplications [50,51]. Using this technique, we identified ten duplications and one deletion, which were between 100 kb and 1 Mb in length (Fig 2A, S4 Table). The robustness of this analytical approach was underscored by detecting the same duplications in M2 siblings (S2 Fig, S4 Table). Next, we employed LUMPY Express [52] to identify structural variants (SVs) in mutant and control plants. This pipeline identified 2,224 variants that included inversions, deletions, duplications, and breakends, which were then filtered for high-confidence events. Breakends were excluded from further analyses because of the challenges of identifying their molecular nature. A high proportion of predicted SVs mapped to Nucleolar Organizer Regions (NOR), plastid genomes, and a small subset of other genomic loci. To test the potential for the induction of SVs in difficult-to-map repetitive regions, we measured read depth at rDNA and telomeres and assessed if etoposide treatment was linked to a higher degree of repeat copy number variation. Although the number of control plants was insufficient to make a definitive statement, we found that the variation in read depth among etoposide-treated lines was comparable to that in control lines, suggesting that etoposide treatment likely did not trigger additional genomic instability in repetitive DNA (Fig 2B). We also found that these SVs were present in control plants, suggesting that they represent mapping artifacts or naturally occurring SVs extant in wild-type plants. Based on these observations, we excluded SVs mapping to these regions from further analyses. To further filter out etoposide-independent SVs, those that were found in four or more etoposide-derived lineages were excluded, although we cannot exclude the possibility that these represent common fragile sites. After applying these stringent filters, we identified 27 unique SVs including 16 deletions and three duplications that ranged between 35 bp and 950 bp in size (S3, S4 Figs, S5 Table). We also identified eight inversions that span between 124 bp and 4 Mb in length (S3, S5 Figs, S5 Table). In sum, the coverage-based approach and LUMPY Express analysis suggest that etoposide treatment induces structural variation in Arabidopsis.

The short-read data was also used to test if etoposide treatment resulted in increased single nucleotide variants (SNVs), or short indels. SNV and short indel analysis identified a comparable number and spectrum of SNVs in etoposide-treated and control lines (Fig 2C, D), suggesting that etoposide did not induce excess SNVs or short indels. Nevertheless, we cannot completely rule out the possibility that some SNVs and short indels may arise due to exposure to etoposide. Therefore efforts by users of etoposide mutagenesis to identify mutations underlying phenotypes of interest should also employ analyses that can accurately identify SNVs and small indels.

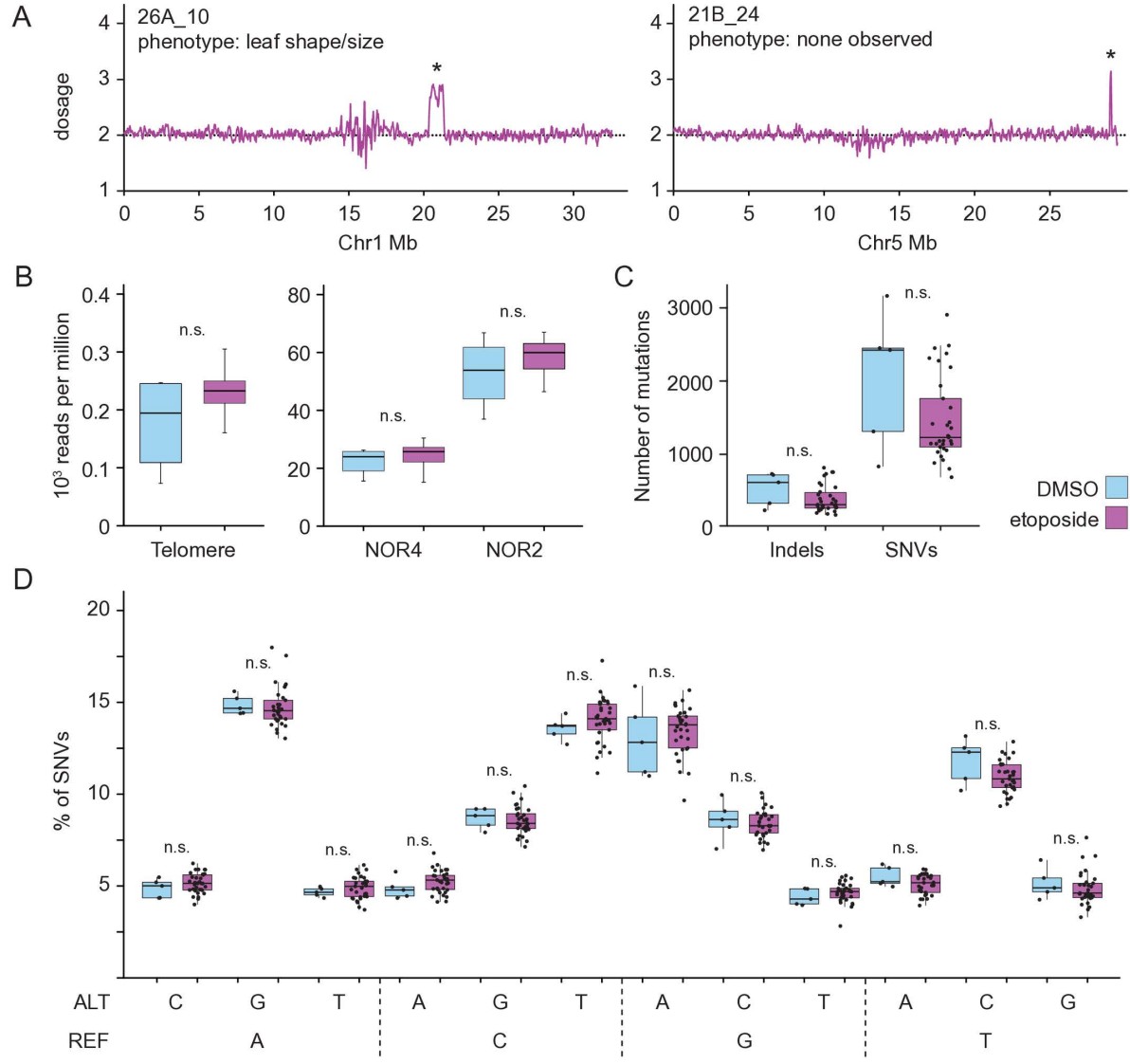

**Fig 2. Impact of etoposide on SNVs and repetitive regions.** (A) Duplications in etoposide-treated lines as assessed by short-read sequencing. Dosage across the indicated chromosome is shown. Chromosomal segments with one extra copy are indicated with *. (B) For Illumina short-read libraries of each control and etoposide-treated plant that we sequenced, total number of reads aligning to the telomere (CCCTAAA), NOR2, and NOR4 were normalized to total numbers of aligned reads in each library. Number of thousand reads, per million sequenced reads, for all control and etoposide treated libraries are represented in this boxplot. Wilcoxon test shows no statistical difference in the median value of read-depth over repetitive regions in control and etoposide-treated plants. (C) Total number of indels and single nucleotide variants (SNVs), genome-wide, identified in etoposide-treated (purple) and control lines (blue). (D) Box-plot describes the composition of nucleotide changes in progeny of plants exposed to etoposide and progeny of plants exposed to DMSO. All SNVs passing quality filters were included here. REF indicates reference allele and ALT represents the alternate SNV. Wilcoxon test was used to determine the significance of differences in SNV numbers between the progeny of etoposide and DMSO treated plant; n.s indicates $p > 0.05$.

## Applications

### Novel recessive and dominant phenotypes can be created by etoposide mutagenesis

To assess the applicability of etoposide mutagenesis in creating novel traits and to establish genotype-phenotype relationships, we closely examined four M1 lines with visible phenotypes: a dwarf line reminiscent of weak

brassinosteroid-insensitive mutants (Fig 1C) [53], referred to as *brassinosteroid-like (BR-like) dwarf* (line 1A); a dwarf line with short internodes (line 34C), or *short-internode dwarf* (Fig 1D); a line with delayed greening (line 13B), termed *virescent* (Fig 1E); and a line with a *variegated* phenotype (line 5A) (Fig 1F). The semi-sterile *BR-like dwarf* line (Fig 1C) exhibited short, thick stems that bore fleshy leaves. In contrast, the *short-internode dwarf* mutant, though short in stature, was fertile, produced the same number of nodes on the primary inflorescence as WT plants (S1H, I Fig), and lacked the fleshy leaf phenotype observed in the *BR-like dwarf* (Fig 1D). In the *virescent* line, juvenile plants exhibited reduced chlorophyll pigmentation, whereas adult plants exhibited normal pigmentation (Fig 1E). *Variegated* mutants displayed light- and temperature-dependent variations in the proportions of green and white sectors on cotyledons and true leaves (Fig 1F).

The *BR-like dwarf, short-internode dwarf,* and *virescent* phenotypes were transmitted from self-fertilized parents to offspring for at least three additional monitored generations, indicating that the phenotype was true-breeding and the underlying mutant genotype was stable (Fig 1B). To assess whether these phenotypes were dominant or recessive, mutant lines were crossed to wild-type. The resultant F1 progeny from the *BR-like dwarf* exhibited intermediate dwarfing, suggesting that the phenotype was incompletely dominant (Fig 1G). The F1 progeny from the *short-internode dwarf* line and the *virescent* line displayed a wild-type phenotype, indicating that these mutant phenotypes were recessive (Fig 1H, I). *Variegated* plants did not flower readily, so this phenotype was maintained as a heterozygous stock for the same number of generations. When self-fertilized, plants from this line either produced all green progeny or one-quarter variegated progeny, indicating that this phenotype was recessive (S6 Fig).

To estimate the number of loci contributing to the *BR-like dwarf*, *short-internode dwarf*, and *virescent* phenotypes, we examined F2 progeny obtained by self-fertilizing the F1 plants. Of F2 progeny of F1 *BR-like dwarf* plants with an intermediate dwarfing phenotype, 23.4% had a wild-type phenotype, 49.6% exhibited an intermediate phenotype, and 27.0% had a severe mutant phenotype (Fig 1G). This result suggested that a single incompletely dominant locus caused the *BR-like dwarf* phenotype (58:123:67 severe:intermediate:wild-type, $H_0 = 1:2:1$ ratio, $\chi^2 = 0.669$; df = 2, $p > 0.5$). The *short-internode dwarf* phenotype was observed in 1/8 of progeny obtained by self-fertilizing F1 plants (28:195 *short-internode dwarf*:wild-type) (Fig 1H). This ratio is consistent with the recessive phenotype being caused by mutations in two linked loci, among other possibilities. For the *virescent* line, 20.9% of the F2 progeny displayed the mutant phenotype (Fig 1I). This is consistent with the phenotype being caused by mutation of a single locus (29:114 *virescent*:wild-type, $H_0 = 1:3$, $\chi^2 = 1.69$; df = 1, $0.25 > p > 0.1$). These observations indicate that etoposide mutagenesis can efficiently create novel recessive and dominant phenotypes caused by alteration to one or more loci.

To further characterize the four phenotypes, we performed mRNA sequencing of rosette leaves for M3 plants of *BR-like dwarf, short-internode dwarf, virescent,* and *variegated* lines (S6 Table). As controls, we included genetic relatives that lacked the phenotype and M3 progeny of DMSO-treated plants. To obtain a broad overview of changes in gene expression, we performed gene set enrichment analysis (GSEA) for biological process gene ontology (GO) on the full ranked list of genes (ranked by $\log_2$ fold change) for each phenotypic line. We also identified differentially expressed genes for each phenotype, defined as genes with an adjusted $p$-value < 0.01 and a $|\log_2(\text{fold change})| > 1$ (S7–S9 Tables).

For the *BR-like dwarf* phenotype, a total of 309 genes were differentially expressed in *BR-like dwarf* plants compared to non-phenotypic plants (114 downregulated and 195 upregulated; Fig 3A, S7 Table). The five most significant GO terms that were overrepresented among upregulated genes were: meristem development, anatomical structure formation involved in morphogenesis, response to auxin, plant organ formation, and post-embryonic plant morphogenesis (S7A Fig). *AS1* was a differentially expressed gene ($\log_2$FC = -2.62, adjusted $p$-value = 6.37 x 10$^{-16}$), along with several genes it is known to interact with either directly or indirectly such as *KNAT1*, *KNAT2*, *KNAT6*, *STM*, *BOP1*, *BOP2*, and *LOB* (Fig 3A, S7 Table) [54–57]. The phenotypes we observed in the *BR-like dwarf* plants were strikingly similar to those described for *as1* mutants, including a compact rosette with lobed and curled leaves [54,58].

In contrast to *BR-like dwarf* plants, *short-internode dwarf* plants showed considerably fewer gene expression changes in leaves. A total of 44 genes (18 down and 26 up) were differentially expressed in *short-internode dwarf* leaves compared

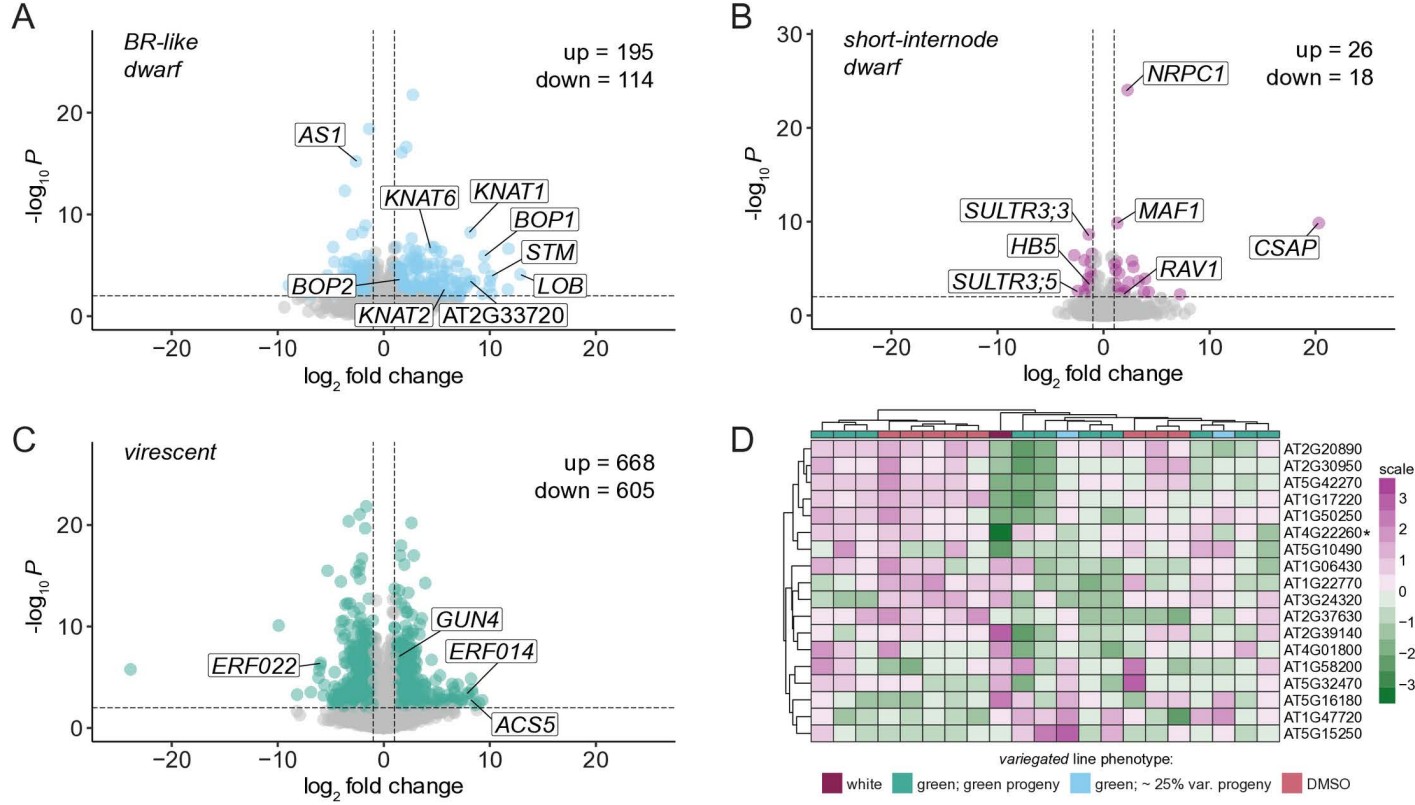

**Fig 3. Etoposide-induced mutants exhibit changes in gene expression.** Volcano plots summarizing differential expression analysis of phenotypic vs. non-phenotypic plants for (A) *BR-like dwarf*, (B) *short-internode dwarf*, and (C) *virescent* mutants. Significantly differentially expressed genes ($|\log_2 FC| > 1$ and adjusted $p$-value < .01) are highlighted in (A) blue, (B) pink, and (C) green. Select genes are labeled in each volcano plot. (D) Heatmap of expression of genes known to be associated with variegation, in the *variegated* line. The most downregulated gene among them, *IMMUTANS*, is annotated with a *.

to control leaves (Fig 3B, S8 Table). GSEA yielded primarily stress-associated GO terms (S7B Fig). Interestingly, both the largest subunit of RNA polymerase III, *NRPC1*, and the repressor of RNA polymerase III, *MAF1*, were upregulated. It is possible that *short-internode dwarf* plants exhibit changes in gene expression in plant organs other than the leaf, or perhaps exhibit a global change in gene expression that cannot be detected with standard mRNA-seq approaches.

Virescent plants had more dysregulated genes than either dwarf line, with a total of 1273 differentially expressed genes (668 up and 605 down; Fig 3C, S9 Table). The most significant GO terms overrepresented amongst genes with increased expression included cell division, plant-type cell wall organization/biogenesis, mitotic cell cycle, microtubule-based process, and meiotic cell cycle (S7C Fig). The most significantly overrepresented GO terms among genes with decreased expression included response to chitin, response to hypoxia, and response to oxygen levels (S7C Fig). Genes related to ethylene synthesis and signaling including *ERF022*, *ERF014*, and *ACS5* were among the most differentially expressed genes (S9 Table). The mis-regulation of these genes might be associated with the *virescent* phenotype, as ethylene is involved in greening of etiolated seedlings after light exposure, and conversely degradation of chlorophyll during leaf senescence [59,60]. Other differentially expressed genes of note include several subunits of photosystem I and II and *GUN4* (S9 Table).

Differential expression analysis could not reliably be conducted for the *variegated* line, as only one variegated plant was evaluated (S6 Table). As an alternative strategy, we examined the expression of 18 genes known to be associated

with variegation (Fig 3D). In the one variegated plant we recovered and performed RNA-seq on, the *IMMUTANS* (*IM*; AT4G22260) gene stood out as having highly reduced expression (Fig 3D). In *variegated* plants we observed sensitivity of the variegation to light and temperature, which is consistent with that displayed in *im* mutants [61,62].

**Long-read sequencing identifies candidate causal mutations**

Long-read sequencing can identify structural variants that are not easily resolved or cannot be identified by short-read sequencing [63]. Using Nanopore sequencing technology, we sequenced the genomes of two M3 plants with the *BR-like dwarf*, *short-internode dwarf,* or *virescent* phenotypes; from each corresponding line an M3 relative that lacked the phenotype; and a single M3 plant from the *variegated* line that was green but produced 25% variegated progeny (S10 Table). To control for structural variation already present in our wild-type Col-0 lab stock compared to the reference Col-0 genome, we also sequenced two M2 plants that were the progeny of M1 plants grown on DMSO and an untreated Col-0 plant. Reads were aligned to the Col-CEN v1.2 reference genome [64] and SVs were called using cuteSV [65,66]. Structural variants present in the DMSO and/or Col-0 controls, present in more than one M1 line, or with a quality less than 20 were discarded. In total we identified 12 SVs: seven deletions (ranging from 37 bp to 176 bp), two insertions (71 bp and 184 bp), two inversions (1.52 Mb and 3.54 Mb), and one translocation (Fig 4A; S11 Table). We also chose to confirm several of the SVs via PCR (S8 Fig): a 184 bp insertion with homology to an intergenic region of chromosome 2 present in the *short-internode dwarf* line (Fig 4B); an 81 bp and a 37 bp deletion present in the *variegated* line (Fig 4C, D); and a 1.52 Mb inversion present on chromosome 2 in the *BR-like dwarf* line (Fig 4E). Together, these data indicate that etoposide treatment generates a range of SV types and sizes.

Finally, we evaluated the relationships between structural variation identified by Nanopore sequencing and gene expression in *BR-like dwarf, short-internode dwarf,* and *variegated* mutants. Both sequenced *BR-like dwarf* plants contain a 1.52 Mb inversion on chromosome 2 that is not present in non-phenotypic control plants of the same line (Fig 4A, E, S11 Table) and we further investigated gene expression surrounding this locus. The 1.52 Mb inversion was significantly enriched for differentially expressed genes, as determined by a permutation test (S9A, B Fig). Examining the inversion breakpoints more closely, the right breakend separated the first 42 bp of the 5'UTR of *ASYMMETRIC LEAVES 1 (AS1;* AT2G37630) from the rest of the gene (Fig 4E). Near the left breakend of the inversion, the expression of an AP2/B3-like transcriptional factor family protein (AT2G33720) was increased in leaves (Figs 3A, 4E, S7 Table). Taken together, these data suggest that the 1.52 Mb inversion resulted in altered *AS1* transcript abundance, influencing leaf patterning in *BR-like dwarf* plants.

Three SVs were common to the two *short-internode dwarf* plants that were sequenced via Nanopore: a 176 bp deletion on chromosome 3, a 3.54 Mb inversion on chromosome 4, and a 184 bp insertion on chromosome 5 (Fig 4A, B, S11 Table). The large inversion on chromosome 4 was not enriched for DEGs, and neither were regions surrounding the deletion on chromosome 3 or the insertion on chromosome 5 (S9C–L Fig). However, since gene expression data was derived from only the leaves of *short-internode dwarf* plants, we cannot exclude the possibility that these structural variants have an impact on gene expression in other plant organs. No structural variants were identified in *virescent* plants utilizing either short- or long-read data. Although it is possible that no structural variation is present in this line, it is also possible that *virescent* plants contain an SV that cannot be detected with our methods due to its complexity or genomic location.

Long-read sequencing of the *variegated* line identified a 37 bp deletion in the third exon of *IM*; this caused a frame-shift resulting in a premature stop codon (Fig 4D). Crossing plants of the *variegated* phenotype to known *immutans* mutants failed to complement the variegated phenotype (32/32 variegated when crossed to *im-spotty* CS3639 [67], 34/34 variegated when crossed to *im-spotty* CS73029 [68], and 42/42 variegated when crossed to *im* CS3157), indicating that the 37 bp deletion in *IM* is indeed the causal mutation for *variegation*.

In summary, our genomic analyses of mutants shows that etoposide treatment can create novel plant traits via the induction of SVs including inversions, deletions, duplications, and translocations.

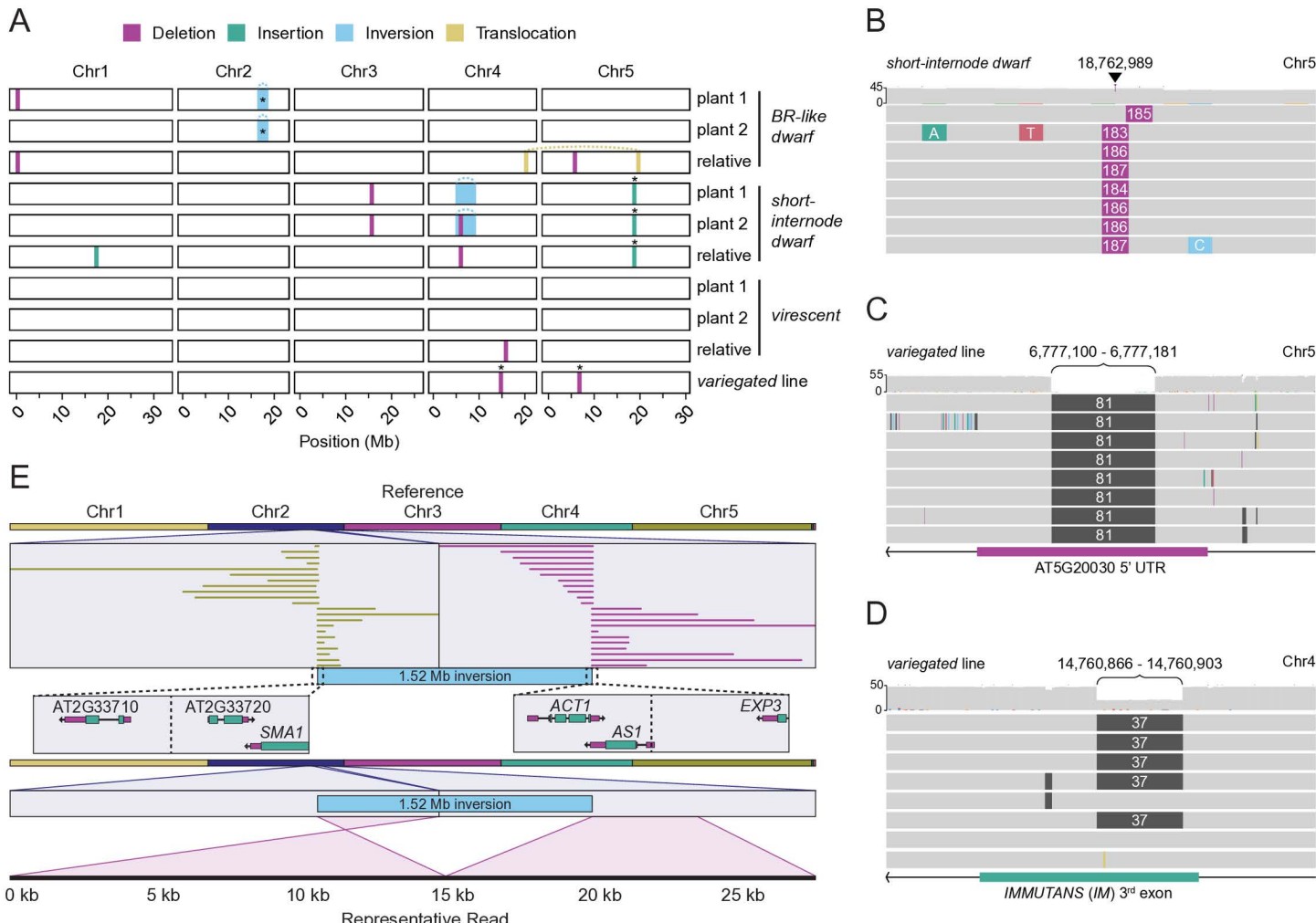

**Fig 4. Whole genome sequencing identifies structural variation in progeny of etoposide treated plants.** (A) Nanopore long-read sequencing identifies structural variation in lines that show the *BR-like dwarf*, *short-internode dwarf*, *virescent*, and *variegated* phenotypes. SVs verified via PCR are indicated with *. Putative causal mutations were identified by sequencing two plants with a phenotype and one relative without a phenotype. Only one mutant plant was sequenced for the *variegated* phenotype. This plant was green but produced ~25% *variegated* progeny. Three PCR-verified mutations in (A) are described in (B-D). (B) A genome browser snapshot with reads identifying a 184 bp insertion on Chr5 in the *short-internode dwarf* line. (C) Reads in the genome browser snapshot indicate an 81 bp deletion on Chr5 in the *variegated* line. (D) Snapshot of long-read alignments to reference *IMMUTANS*, showing a 37 bp deletion in the third exon. Deleted sequence shown in grey and coverage is shown above read alignments. The deletion is heterozygous in the sequenced plant, which was green and produced ~25% *variegated* progeny. (E) A ribbon plot with alignments indicating a 1.52 Mb chromosome 2 inversion in the *BR-like dwarf* line (top) and a detailed view of a representative read spanning the right breakend. The right breakend is in the 5'UTR of the *AS1* gene.

## Discussion

### Etoposide treatment is an efficient method to induce structural variation

We describe a simple and convenient technique that enables researchers or breeders with limited resources to induce structural variation. We found that early exposure of *A. thaliana* to the Topo II inhibitor etoposide induces heritable, large-effect mutations at high frequency. Examination of M2 plants found that 24/37 mutant lineages had at least one out of six assessed phenotypes. By short- and long-read genomic sequencing of mutagenized lines—including those that displayed

novel phenotypes—we identified the full spectrum of structural variations. Analysis of short-read data identified 2.53 SVs per mutagenized M1 line whereas long-read data identified 3.25 SVs per mutagenized M1 line, with one event being detected in both short and long-read data. The lower number of SVs identified by short-read data is consistent with the poorer performance of short-read SV callers relative to callers using long-reads [63]. Our data suggest that mutation rate and type induced by etoposide exposure is comparable to or out-performs mutant populations generated by irradiation. Analysis of Arabidopsis seeds irradiated with gamma rays or carbon ions showed mutations occurring at a rate of 0.8-1.6 SVs per mutant lineage [69]. While the exact efficacy of irradiation or etoposide-based mutagenesis can vary by experiment, currently available data suggests that etoposide treatment exceeds irradiation by producing more than 2.5 events per mutagenized plant. A further increase in the number of mutations per M1 parent could perhaps be attained either by prolonging exposure to etoposide beyond two weeks or by germinating seeds on media containing a cocktail of etoposide and other drugs that induce genomic instability such as topoisomerase I inhibitors [70] or the DNA damage-inducing agent cisplatin [71].

Analysis of all detected SVs suggests that etoposide disproportionately impacts genes, although our analytical approaches cannot conclusively identify all SVs in heterochromatin. The SVs that we identified disrupt one or more genes simultaneously or create novel cis-regulatory regions. The effect of this genic bias is striking; even in our relatively small mutant population, multiple potential loss-of-function mutations were identified in a single line. This suggests that etoposide treatment can be used to create novel polygenic traits.

What explains this bias for mutations in genes? Topoisomerase II has been shown to associate with actively transcribed regions and open chromatin regions [42]. This association with actively transcribed regions suggests a future method to enrich for mutations in condition-specific genetic pathways by simultaneously exposing seeds to etoposide and another condition. For example, etoposide might promote mutagenesis of immunity genes whose transcription is triggered when seeds are germinated in the presence of microbial pathogens. In another scenario, seeds germinated in the presence of a hormone and etoposide might accumulate mutations in genes that respond to said hormone.

The inversions and translocations generated by this technique might also disrupt meiosis and distort segregation, but this possibility has not been tested. Etoposide also created duplications and deletions that could potentially alter gene dosage over megabase scale regions. In allopolyploids, such segmental deletions can provide a deficiency chromosome that can be used to assess the relative contribution of homeologs to different traits.

## Identifying causal mutations

How do we identify genetic changes that are causative for phenotypes of interest? Future studies will accelerate candidate-gene discovery by employing structural-variant callers, *de novo* genome assembly-based approaches, and RNA-seq based mapping [63]. Although our study did not aim to use RNA-seq to identify causative mutations, it provides an example of how RNA-seq data in tandem with genome sequencing can help shortlist potential causal mutations. In cases where a potential causative variant is not obvious, these strategies can be combined with traditional genetic mapping approaches. However, mapping-by-sequencing approaches might not be easily applicable to some classes of mutants—for example, inversions or translocations can suppress recombination and reduce the efficacy of mapping-by-sequencing.

In our current study, sequencing and close analysis of the genomes and transcriptomes of *BR-like dwarf, short-internode dwarf, virescent,* and *variegated* plants allowed us to link two of these phenotypes to strong candidate mutations. The *BR-like dwarf* phenotype is likely caused by an inversion that disrupts the expression of *ASYMMETRIC LEAVES 1* and the *variegated* phenotype is caused by a 37 bp deletion in *IMMUTANS.* Strikingly, 60 years ago, the *asymmetric leaves* and *immutans* phenotypes were amongst the first X-ray induced mutations isolated in *A. thaliana* by Redei [72,73].

## Etoposide mutagenesis offers several advantages compared to other methods

At present, the most common way to induce structural variation in plant species is irradiation-based methods. Radiation sources are heavily regulated and access to them is a bottleneck in the creation of structural variant libraries [26]. The

method described here eliminates this bottleneck by allowing researchers to generate structural variant libraries in their home labs without the need to identify a radiation source or send material elsewhere. Additionally, application of irradiation can be tedious, sometimes requiring collecting and irradiating dried pollen [22]. Etoposide treatment is less laborious, and may be used with any species in which seeds can be germinated and grown for a brief period of time on MS media.

Prior to the method described here, alternatives to radiation-based mutagenesis (such as the extended TAQing approach [37]) required transformation of plant species; however the commercial varieties of many crops are not easily transformed. The easy-to-use chemical mutagenesis approach we describe provides a valuable alternative to irradiation-based methods for difficult to transform varieties and/or species. Our method generates mutant libraries that can be used directly for research or breeding, or provide targeting information for CRISPR-based tools that have recently created megabase scale SVs in crop species [74–77]. Overall, etoposide-mutagenesis provides plant biologists and breeders with a new tool that can replace irradiation and reduce barriers to the creation of structural variation.

## Supporting information

**S1 Text. Detailed description of materials and methods.**
(DOCX)

**S1 Fig. Additional phenotyping of progeny of etoposide-treated plants.**
(PDF)

**S2 Fig. Duplications detected in the 26A lineage.**
(PDF)

**S3 Fig. Structural variants identified by LUMPY Express in short-read data.**
(PDF)

**S4 Fig. Examples of deletions in an etoposide-treated line identified by LUMPY Express using short reads.**
(PDF)

**S5 Fig. An example of an intrachromosomal inversion in an etoposide-treated line identified by LUMPY Express using short reads.**
(PDF)

**S6 Fig. Segregation of the *variegated* phenotype from green parents.**
(PDF)

**S7 Fig. GO term enrichment for genes differentially expressed between plants exhibiting a phenotype and those without a phenotype.**
(PDF)

**S8 Fig. PCR verification of four structural variants identified via long-read sequencing.**
(PDF)

**S9 Fig. Permutation tests to determine if structural variants are associated with location of differentially expressed genes.**
(PDF)

**S1 Table. Phenotypes observed in M2 progeny of etoposide-treated plants.**
(XLSX)

**S2 Table. List of samples sequenced with Illumina.**
(XLSX)

**S3 Table. Relationship between the sequenced lines, treatments, and nature of sequencing.**
(XLSX)

**S4 Table. Deletions and duplications identified by read coverage analysis.**
(XLSX)

**S5 Table. Structural variants detected by LUMPY Express using short-read sequencing.**
(XLSX)

**S6 Table. List of RNA-Seq libraries.**
(XLSX)

**S7 Table. DESeq2 output identifying mis-regulated genes in *BR-like dwarf.***
(XLSX)

**S8 Table. DESeq2 output identifying mis-regulated genes in *short-internode dwarf.***
(XLSX)

**S9 Table. DESeq2 output identifying mis-regulated genes in *virescent*.**
(XLSX)

**S10 Table. List of samples sequenced with Nanopore.**
(XLSX)

**S11 Table. Structural variants identified using Nanopore long-read data.**
(XLSX)

## Acknowledgments

We thank Aysha Hussain, Jaza Alam, and Lingfeng Shi for assistance with phenotyping, Dyna Louis for assistance with verifying structural variants, and Xinlei Gao for assistance with statistical testing.

## Author contributions

**Conceptualization:** Mary Gehring, Prasad R. V. Satyaki.

**Data curation:** Lindsey L. Bechen, Naiyara Ahsan, Prasad R. V. Satyaki.

**Formal analysis:** Lindsey L. Bechen, Naiyara Ahsan, Alefiyah Bahrainwala, Prasad R. V. Satyaki.

**Funding acquisition:** Mary Gehring.

**Investigation:** Lindsey L. Bechen, Naiyara Ahsan, Alefiyah Bahrainwala, Prasad R. V. Satyaki.

**Methodology:** Lindsey L. Bechen, Mary Gehring, Prasad R. V. Satyaki.

**Project administration:** Mary Gehring.

**Supervision:** Mary Gehring, Prasad R. V. Satyaki.

**Visualization:** Lindsey L. Bechen, Naiyara Ahsan, Alefiyah Bahrainwala, Mary Gehring, Prasad R. V. Satyaki.

**Writing – original draft:** Lindsey L. Bechen, Naiyara Ahsan, Mary Gehring, Prasad R. V. Satyaki.

**Writing – review & editing:** Lindsey L. Bechen, Mary Gehring, Prasad R. V. Satyaki.

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
