## [Editor Report · Decision Letter 0]

18 Jun 2025

PGENETICS-D-25-00577

A simple method to efficiently generate structural variation in plants

PLOS Genetics

Dear Dr. Satyaki,

Thank you for submitting your manuscript to PLOS Genetics. Your work raised significant interest among the editors, and there was a consensus that the technical quality of the study is high. However, some concerns were expressed regarding the novelty of the findings in their current form. Given the strong methodological component of your study, we believe it holds potential for publication as a Methods paper.  Therefore, we invite you to submit a revised version of the manuscript that addresses the points raised during the review process.  Please see the Additional Editor Comments at the end of this email for a summary of our suggested revisions.

Please submit your revised manuscript within 60 days (**August 17, 2025)** . If you will need more time than this to complete your revisions, please reply to this message or contact the journal office at plosgenetics@plos.org at any time. Please include the following items when submitting your revised manuscript:

We look forward to receiving your revised manuscript.

Kind regards,

Mathilde Grelon

Academic Editor

PLOS Genetics

Aimée Dudley

Editor-in-Chief

PLOS Genetics

Anne Goriely

Editor-in-Chief

PLOS Genetics

**Additional Editor Comments (if provided):**

Dear Dr Satyaki,

Thank you for submitting your manuscript to PLOS Genetics. Your work raised significant interest among the editors, and there was a consensus that the technical quality of the study is high. However, some concerns were expressed regarding the novelty of the findings in their current form.

Given the strong methodological component of your study, we believe it holds potential for publication as a Methods paper. To move forward, we would like to invite you to submit a thoroughly revised version of your manuscript that meets the criteria for a Methods article.

These criteria include:

- A clear and detailed description of the methodology,

- Demonstration of the method’s utility and reproducibility,

- Explanation of how the method advances or differs from existing approaches,

- Relevance and applicability of the method to the research community.

As the method for generating SVs is already common in non-plant systems, it would be important to highlight how your approach brings added value to the plant research community, and what specific advantages or innovations it offers in comparison to existing methodologies.

We look forward to receiving a revised version of your manuscript tailored to these considerations.

Sincerely yours

Mathilde Grelon

**Journal Requirements:**

**Reviewers' comments:**

**Figure resubmission:**
---

## [Decision Letter · Decision Letter 1]

8 Oct 2025

PGENETICS-D-25-00577R1

A simple method to efficiently generate structural variation in plants

PLOS Genetics

Dear Dr. Satyaki,

Thank you for submitting your manuscript to PLOS Genetics. The reviewers appreciated the work but had some minor suggestions that could improve the manuscript. Therefore, we invite you to submit a revised version of the manuscript that addresses the points raised during the review process. Please note that for manuscripts requiring minor revisions,  the editor usually reviews the revised manuscript and response to reviewers, possibly in consultation with specific reviewers.  We make every effort to shorten time to publication. 

Please submit your revised manuscript within 30 days. If you will need more time than this to complete your revisions, please reply to this message or contact the journal office at plosgenetics@plos.org. Please include the following items when submitting your revised manuscript:

We look forward to receiving your revised manuscript.

Kind regards,

Mathilde Grelon

Academic Editor

PLOS Genetics

Aimée Dudley

Editor-in-Chief

PLOS Genetics

Aimée Dudley

Editor-in-Chief

PLOS Genetics

Anne Goriely

Editor-in-Chief

PLOS Genetics

**Additional Editor Comments (if provided):**

Reviewer #1:

Reviewer #2:

Reviewer #3:

**Journal Requirements:**

**Reviewers' comments:**

Reviewer's Responses to Questions

**Comments to the Authors:**

Reviewer #1: The revision satisfies my criticism.

Reviewer #2: I have to say that I was not involved in reviewing the first version of the paper. I read the revised manuscript as well as the rebuttal letter of the authors with interest. Using etoposide the authors are able to induce a larger number of structural variations in Arabidopsis , mainly insertion and deletions, which they also characterize by long read sequencing. The authors were able to correlate SVs with specific phenotypes, however not all SVs are correlated with phenotypic changes. These are interesting data and I regard them interesting for a large audience. Nevertheless, I feel that the authors are a bit too positive in the discussion comparing their results with irradiation mutagenesis. Moreover, in the last years the induction of directed Mb-sized SVs especially inversions by CRISPR/Cas has been achieved in a row of plant species and I think that this fact should be mentioned in the discussion, too

Reviewer #3: This manuscript, “A simple method to efficiently generate structural variation in plants” by Bechen et al. presents an innovative approach to plant mutagenesis through chemical treatment with the topoisomerase II inhibitor etoposide. The authors show that, unlike commonly used chemical mutagens (e.g., EMS, sodium azide) that mainly induce SNPs, etoposide efficiently generates large structural variants (e.g., indels, inversions, duplications, and translocations) while maintaining wild-type-like SNV frequencies. This method represents a promising alternative to irradiation, offering a more accessible method for inducing complex mutations relevant for plant biology and breeding.

The study combines phenotypic characterization, whole-genome sequencing (both short and long reads), and transcriptome analysis to identify structural variants and potential causal mutations. The manuscript is clearly written, well-illustrated, and supported by thorough experimental work that strongly supports authors conclusions.

Overall, the manuscript introduces a novel, valuable and broadly applicable tool for generating genomic structural variation in plants, with strong potential to impact both fundamental research and applied breeding.

I only have a couple of minor comments on the data and interpretation:

- While the method shows promise, its applicability may be limited by the need for sustained exposure to the inhibitor, which may be less practical in certain contexts compared to radiation. My recollection is that etoposide is relatively unstable in solution and light sensitive. I therefore wonder whether etoposide remains active after several days (up to 2 weeks)? This could explain the varied efficiency of etoposide treatment reported by the authors. Did the authors attempted repeated or renewed etoposide treatments after for example 1 week (e.g. by adding fresh etoposide in the medium if transferring to new plates is tedious)? Or by treating few days after germination? (Since treatment requires active DNA replication, applying it directly to seeds may not be optimal).

Minor point : Cytotoxic drugs usually require neutralization for safe disposal. However, there is no mention of such procedure in the method. Was any neutralization procedure applied that is worth mentioning?

- Line 214- 218 : The text gives impression that DSBs leads to SNV. However, my understanding is that while a DSB can theoretically be repaired in a way that leads to just a single-base substitution, this is not the most common outcome. Instead, DSB repair most often results in small insertions or deletions due to non-homologous end joining NHEJ and/or Microhomology-mediated end joining, or, as described by the authors in the manuscript, larger structural variants (SVs) such as inversions, duplications, or translocations.

All events observed and described by the authors are > 35 bp. Have the authors checked for small Indels (2-30 bp) typically associated with end-joining ? (Of note, cells lacking NHEJ factors are usually hypersensitive to etoposide).

- Line 297-303 : Do the authors have any comments on the upregulation of meiotic genes in Virescent plant leaves? Are these plants later affected in fertility?

- Table S7-to S9 provide the complete RNAseq data. However, the authors might also consider including a list containing only the few deregulated genes, which would be more straightforward to interpret for the reader.

**Have all data underlying the figures and results presented in the manuscript been provided?**

Reviewer #1: Yes

Reviewer #2: Yes

Reviewer #3: Yes

PLOS authors have the option to publish the peer review history of their article (what does this mean? ). If published, this will include your full peer review and any attached files.

**Do you want your identity to be public for this peer review?** For information about this choice, including consent withdrawal, please see our Privacy Policy .

Reviewer #1: No

Reviewer #2: No

Reviewer #3: No

**Figure resubmission:**
---

## [Editor Report · Decision Letter 2]

13 Nov 2025

PGENETICS-D-25-00577R2

A simple method to efficiently generate structural variation in plants

PLOS Genetics

Dear Drs Gehring and Satyaki,

Thank you very much for this revised version of your manuscript. It is almost ready for PLOS Genetics's publication but we would like to ask you to add details in your answer to Reviewer2/comment 2: please add the **relevant references** to support the added sentence: " Our method will generate mutant libraries that can be used directly for research or breeding, or provide targeting information for CRISPR-based tools that have recently created mega-base scale SVs in crop species **(REFS)** "

 Therefore, we invite you to submit a revised version of the manuscript that addresses the points raised during the review process.

Please submit your revised manuscript as soon as you can ( before the 12/12/25). If you will need more time than this to complete your revisions, please reply to this message or contact the journal office at plosgenetics@plos.org. Please include the following items when submitting your revised manuscript:

We look forward to receiving your revised manuscript.

Kind regards,

Mathilde Grelon

Academic Editor

PLOS Genetics

Aimée Dudley

Editor-in-Chief

PLOS Genetics

Aimée Dudley

Editor-in-Chief

PLOS Genetics

Anne Goriely

Editor-in-Chief

PLOS Genetics

**Additional Editor Comments (if provided):**

**Journal Requirements:**

**Reviewers' comments:**

**Figure resubmission:**
---

## [Editor Report · Decision Letter 3]

2 Dec 2025

Dear Dr Satyaki,

We are pleased to inform you that your manuscript entitled "A simple method to efficiently generate structural variation in plants" has been editorially accepted for publication in PLOS Genetics. Congratulations!

Yours sincerely,

Mathilde Grelon

Academic Editor

PLOS Genetics

Aimée Dudley

Editor-in-Chief

PLOS Genetics

Aimée Dudley

Editor-in-Chief

PLOS Genetics

Anne Goriely

Editor-in-Chief

PLOS Genetics

BlueSky: @plos.bsky.social

Comments from the reviewers (if applicable):

**Data Deposition**

http://datadryad.org/submit?journalID=pgenetics&manu=PGENETICS-D-25-00577R3

**Press Queries**

---

## [Editor Report · Acceptance letter]

PGENETICS-D-25-00577R3

A simple method to efficiently generate structural variation in plants

Dear Dr Gehring,

We are pleased to inform you that your manuscript entitled " 

A simple method to efficiently generate structural variation in plants" has been formally accepted for publication in PLOS Genetics! Your manuscript is now with our production department and you will be notified of the publication date in due course.

With kind regards,

Zsofia Freund

PLOS Genetics

On behalf of:
